# *Juniperus horizontalis* Moench: Chemical Composition, Herbicidal and Insecticidal Activities of Its Essential Oil and of Its Main Component, Sabinene

**DOI:** 10.3390/molecules27238408

**Published:** 2022-12-01

**Authors:** Daniela Gruľová, Beáta Baranová, Vincent Sedlák, Laura De Martino, Valtcho D. Zheljazkov, Mária Konečná, Janka Poráčová, Lucia Caputo, Vincenzo De Feo

**Affiliations:** 1Department of Ecology, Faculty of Humanities and Natural Sciences, University of Presov, Ul. 17. Novembra 1, 08001 Prešov, Slovakia; 2Department of Biology, Faculty of Humanities and Natural Sciences, University of Presov, Ul. 17. Novembra 1, 08001 Prešov, Slovakia; 3Department of Pharmacy, University of Salerno, Via Giovanni Paolo II, 132, 84084 Fisciano, Italy; 4Department of Crop and Soil Science, Oregon State University, Corvallis, OR 97331, USA; 5Research National Council, Institute of Food Science, 831000 Avellino, Italy

**Keywords:** GC-MS, crops, weeds, phytotoxic activity, larvicidal activity, *Tenebrio molitor* L., *Tubifex tubifex* Müller 1774, Chironomus aprillinus Meigen 1830

## Abstract

The composition, herbicidal, larvicidal, and toxic activities of *Juniperus horizontalis* Moench essential oil and sabinene, its main component, were evaluated. The seed germination percentage and root length of eight different plant species (crops and weeds) were measured for in vitro herbicidal activity tests. Different doses (100, 50, 10, 5, 1, 0.5 µg/mL) of the samples were applied to seeds for 120 h. The same doses were applied to test the toxicity of the samples on *Tubifex tubifex* (sludge worm) and *Chironomus aprilinus* (blood worm). Four doses (435, 652.5, 870, and 1740) of samples were in a larvicidal test on *Tenebrio molitor* L. (mealworm), and bioassays were checked after 24 and 48 h. The analysis of the oil allowed for the identification of dominant components as sabinene (38.7%), α-pinene (10.0%), elemol (8.6%), γ-terpinene (8.3%), limonene (7.8%) and α-thujene (5.3%). The results showed that the effect of oil on root length inhibition was significant in all crop species. The doses which affected root growth were not toxic to *Tubifex tubifex* and *Chironomus aprilinus*. Finally, the obtained results in the larvicidal bioassay indicate that the potential of *J. horizontalis* in yellow mealworm survivorship limitation can be a starting point for future research.

## 1. Introduction

Coniferous trees and shrubs, known as junipers, belong to the *Juniperus* genus (Cupressaceae). Approximately 50 to 67 species of this genus are present in the Northern Hemisphere, from the Arctic to South Africa and between western Europe and Tibet [1]. Previous and recent studies focus on the ecological, morphological, chemical, medicinal, and molecular characteristics of different *Juniperus* species [1,2,3]. *Juniperus horizontalis* Moench, known as creeping juniper or creeping cedar, is a small shrub native to North America: it grows in Canada, in the United States, in Alaska, and from Montana to Maine, between south of Wyoming and northern Illinois [4]. It is 10 to 30 cm tall and spreads a few meters wide. This species is closely related to *J. virginiana* L. (eastern juniper) and to *J. scopulorum* Sarg. (rocky mountain juniper), with which it often hybridizes in southern Canada. Several varieties were bred and cultivated in gardens and around the houses [4]. Few papers report the essential oil composition in relation to chemotaxonomic studies of this genus [5,6]. The biological activities of the juniper genus depend on its secondary metabolites released to the surrounding. Other recent publications refer to its biological activity [1,4,7]. The EO of *J. horizontalis* is reported for its potential phytotoxic effect on lettuce [8].

In fact, secondary metabolites possess the strength to affect ecosystem function and structure [9]; they exert a role in the control of plant diversity [10]. The plants use exudation, decomposition, vaporization, or leaching to introduce numerous substances into the environment, based on their physicochemical characteristics and the specific organ of production and/or accumulation [9].

Although many essential oils of the different *Juniperus* species have similarities, their components do not necessarily produce the same biological effects. The chemical markers responsible for the targeted activity, in addition to the chemical constituents, are useful for discriminating between different essential oils [11]. Sabinene, one of the most abundant components in *Juniperus* essential oils, has previously been reported in the literature for its biological activity [1].

The unmanaged weeds cause a greater reduction in crop yields than the presence of any other agricultural pest [12]. In the context of sustainable agriculture, the need to control weeds has led to the rediscovery of the chemical interactions between living organisms within the ecosystem. The need to design new pesticides in the context of environmentally sustainable agriculture requires in-depth knowledge of the often neglected structural and chemical characteristics of the substances. Many of these substances have been used in agricultural practices for their herbicidal effect or for their ability to regulate plant growth [13]. The available natural herbicides have little or no selectivity and must be applied in relatively large quantities in comparison with synthetic herbicides. Furthermore, little scientific literature is available on the use and environmental impact of natural products in organic agriculture [12]. The identification of new and suitable inhibitors is important for the development of new herbicidal substances with higher agronomic capacity, lower environmental impact, and fewer resistance problems [14]. Furthermore, in order to meet the regulatory requirements in the design and subsequent development of a new pesticide, a critical factor to evaluate is the presence of any adverse effects on the environment and on non-target organisms [15]. The use of volatile substances in the suppression of unwanted plants during the seedling stage could be a good opportunity to reduce the employment of industrial agrochemicals [16].

Different pests cause various damages to crops or agricultural products as well as to human or animal health. The essential oils of several plants have also been identified as natural insecticides. Several studies have provided evidence of larvicidal activity on *Artemia salina*, *Aedes aegypti*, *Anophelus sinensis* [17,18,19], and others. In particular, *Tenebrio molitor* Linnaeus, the mealworm beetle (Coleoptera: Tenebrionidae), is a parasite of stored products, such as starches and pasta: this insect is able to infest the broken grains of *Zea mays* (L.) and *Triticum aestivum* L. The presence of body fragments and/or feces in stored grain and bran can lead to a loss of food quality [20].

The aims of this research are to identify the composition of EO of *J. horizontalis* and to evaluate the possible phytotoxic effects on plant germination, the larvicidal activity on mealworms, and the toxic activity on sludge worms and blood worms. In addition, sabinene, the main component of the oil, was tested. The safety profile of the oil and sabinene is also checked.

The knowledge about the different biological effects of *Juniperus horizontalis* is missing in the literature. In the current research, crops and weed species have been selected according to availability in our locality. The hypothesis is that the essential oil of *J. horizontalis*, with the dominant compound sabinene, has a phytotoxic effect on plant species, generally identified as weeds, as well as some larvicidal effect.

## 2. Results

### 2.1. Identification of J. horizontalis Essential Oil Constituents

The constituents of *J. horizontalis* essential oil are shown in Table 1. Twenty-nine components were identified, representing 93.4% of the total oil. Monoterpene hydrocarbons were the most abundant compounds (71.0%), followed by oxygenated sesquiterpenes (12.8%), oxygenated monoterpenes (5.1%), and sesquiterpene hydrocarbons (4.5%). Sabinene (38.7%) was the most abundant component; α-pinene (10.0%), elemol (8.6%), γ-terpinene (8.3%), limonene (7.8%) and α-thujene (5.3%) were also present in appreciable amounts.

### 2.2. Influence of J. horizontalis Essential Oil on Seed Germination and on Root Length

Table 2 reported the influence of the *J. horizontalis* EO on seed germination. Within the group of crops, the EO showed the strongest antigerminative effect on *L. sativum* and the lowest on *T. aestivum*: no antigerminative effect of *J. horizontalis* oil was significant on treated seeds. Neither was there any linear relationship nor correlation between EO dose and the percentage of seeds germinated. Only the highest dose of EO, 100 µg/mL, caused a significant decrease in *T. aestivum* germination in comparison to control seeds. Consequently, a significant negative correlation was observed between wheat germination and *J. horizontalis* EO dose (Figure 1). In the group of weeds, no significant effect of the oil on Portulaca oleracea seed germination was observed. On the other hand, the percentage of germinated seeds of *Lolium perenne* and *Barbarea vulgaris* differed significantly from the controls at the doses of 10 µg/mL (*p* ≤ 0.01) and 100 µg/mL (*p* ≤ 0.05), respectively (Figure 2). Similarly, *B. vulgaris* showed a significant negative linear relationship (*p* ≤ 0.05) and a significant negative correlation (*p* ≤ 0.01) between germination percentage and the applied juniper EO dose. The germination percentage of *Trifolium pratense* was significantly lower than the control at doses of 50 µg/mL (*p* ≤ 0.05), 5 µg/mL (*p* ≤ 0.01), and 1 µg/mL (*p* ≤ 0.05) (Figure 3). However, no significant linear relationship or correlation was observed between germination percentage and *J. horizontalis* EO dose.

The impact of *J. horizontalis* EO was greatest on root growth: in each species of the crop group, the root length of seeds treated with all doses of oil was significantly shorter than the root length of control seeds (Table 3). However, no linear relationship and no correlation between root growth and the applied juniper EO dose were observed. The strongest phytotoxic effect was registered with the highest dose of 100 µg/mL. In the weed group, seeds of *T. pratense* and *P. oleracea* were the most resistant: the root length of both seeds was significantly (*p* ≤ 0.05) longer than the length of the control seeds at all doses employed of EO, with the exception of the dose 1 µg/mL of P. oleracea. In *B. vulgaris*, the roots of seeds treated with doses of 100 and 1 µg/mL of *J. horizontalis* EO. were significantly (*p* ≤ 0.05) shorter than the controls. In non-weed species, a linear relationship or correlation between root growth and *J. horizontalis* EO doses was observed. The obtained results showed that the juniper EO mainly influenced the length of the roots of the crop seeds; on weed seeds, juniper EO had a more noticeable effect on germination percentage.

### 2.3. Influence of Sabinene on Seed Germination and on Root Length

Table 4 reported the results of the influence of sabinene, the dominant compound in *J. horizontalis* EO, on seed germination. Sabinene didn’t show any antigerminative activity against *S. alba* and *L. sativum*: the compound, at doses of 100 µg/mL and 5 µg/mL, had an antigerminative effect only on *H. vulgare* seeds. Conversely, sabinene, at the doses of 50 µg/mL, 10 µg/mL, and 5 µg/mL, showed a stimulating effect on the germination of *T. aestivum*, compared to the control. No differences were recorded in the antigerminative effect of doses of sabinene applied to seeds of *T. aestivum*, *H. vulgare,* or *L. sativum*. Furthermore, no linear relationship or correlation was revealed between sabinene doses and the percentage of germinated seeds. Regarding *B. vulgaris*, *P. oleracea,* and *T. pratense*, no differences in the antigerminative activity between control and sabinene were observed at different doses used. Concerning *L. perenne*, the doses of 100 µg/mL, 5 µg/mL, and 0.5 µg/mL of sabinene showed significant antigerminative activity. There was no linear relationship or correlation between the sabinene dose and the percentage of germinated seeds.

The effect of sabinene on root growth was reported in Table 5. The compound had no effect on the root growth of *H. vulgare* and *S. alba*. For *T. aestivum*, the lowest doses of sabinene had a significantly stimulating effect on root growth. On the other hand, after the application of the highest dose, an inhibitory effect of the root growth was recorded compared to the control. A significant negative linear dependence between sabinene doses and *T. aestivum* root length was also observed (Figure 4). The doses of 50 µg/mL, 10 µg/mL, and 5 µg/mL showed a stimulating effect on the root growth of *L. sativum* compared to the control. On the other hand, at a dose of 1 µg/mL sabinene, *L. sativum* showed significantly shorter roots compared to doses of 50 µg/mL, 10 µg/mL, and 5 µg/mL (Figure 5). Sabinene showed a significant phytotoxic effect on *L. perenne* at all doses, except at the lowest dose of 0.5 µg/mL; the strongest impact was observed at the highest dose used. Linear dose dependence and a negative correlation between sabinene dose and *L. perenne* root length were also observed. The opposite effect was observed in the seeds of P. oleracea, on which the application of sabinene had a stimulating effect: in fact, at almost all doses used, the length of the roots was significantly greater than that of the control seeds. A similar stimulating effect was observed in *T. pratense*: however, a significant difference between treated and control seeds was observed only at a dose of 50 µg/mL. The roots of the treated seeds of *B. vulgaris* were shorter than the control, with no significant differences; a significant dose dependence and a negative correlation between sabinene dose and *B. vulgaris* root length were also observed.

### 2.4. Larvicidal Activity and Toxicity of J. horizontalis and of Sabinene

The first death of larvae, as well as sign of necrosis, were observed 24 h after application. A significant (*p* = 0.013) linear dependence between juniper EO dose and yellow mealworm mortality was observed. LD_50_ was 860 µg Insect^−1^ (685–1080), and LD_90_ was 1300 µg Insect^−1^ (1640–2060). Sabinene was not active.

The mortality of *T. molitor* larvae was reported in Table 6.

At all used doses of *J. horizontalis* EO and of sabinene, no mortality from *Tubifex tubifex* was observed. Even prolonged exposure did not cause any sludge worm mortality.

## 3. Discussion

### 3.1. Constituents of J. horizontalis Essential Oil

In *Juniperus horizontalis* EO, the hydrocarbons monoterpenes sabinene, α-pinene, limonene, γ-terpinene, and oxygenated sesquiterpene elemol were the main constituents. The oil was obtained from plants harvested in Wyoming, in the Bighorn Mountains, Canada, with the same origin as other essential oils reported in the literature. In 1961, one of the first reports [21] on *J. horizontalis* EO reported hydrocarbon esters, mixed ketones, and alcohols as main components. In a later paper [5], sabinene (36.5%), limonene (17.5%), terpinen-4-ol (4.6%), elemol (3.8%), and α-pinene (1.6%) were identified as main components in the EO hydrodistilled from branches of *J. horizontalis*. Sabinene was identified as the main component in most cases, in percentages ranging from 46 to 61%, followed by limonene (2.7–7.0%), terpinen-4-ol (3.9–12.5%), elemol (2.0–5.6%), α-pinene (3.6–6.1%) [4]. Also, in this case, the obtained results were in agreement with the literature. In the EOs distilled from different cultivars of *J. horizontalis*, the main components were found in different percentages [6]. The chemical compositions of *J. horizontalis* EOs from Egypt and Turkey were also studied: linalool (33.8%), p-cymene (23.2%), γ-terpinene (8.7%), trans sabinene hydrate (8.6%), and limonene (1.4%) resulted as the main constituents in samples collected in Turkey [1]; in an Egyptian sample, bornyl acetate (23.2%), sabinene (19.5%), 4-terpinenol (8.5%), and α-pinene (7.1%) were the more abundant compounds [7]. The EO sample reported here had a chemical composition very similar to that of *juniper* EOs from North American plants: the composition of EOs was normally influenced by internal (genetics) and external factors [22]. Sabinene, the dominant compound in *J. horizontalis* EO, is a bicyclic unsaturated monoterpene [23], produced by several species [24] and present in many EOs [25], also distilled from different *Juniperus* species, *Juniperus sabina* L. (savin juniper), *J. foetidissima* Willd. (foetid juniper), *J. scopulorum* L. [26,27]. It showed antifungal, antimicrobial, antioxidant, cytotoxic, herbicidal, and anti-inflammatory properties [4,26,27]. Also, α-pinene, β-pinene, myrcene, limonene, and β-caryophyllene were identified as major components in various plant species [1,26].

### 3.2. Effect of the EO and of Sabinene on Seed Germination and on Root Length

*Juniperus horizontalis* essential oil showed to have a variable degree of antigerminati ve potential. A study of two different *Juniperus* species (*J. sabina* L. and *J. excelsa* Bieb.) was recently conducted for evaluation of their phytotoxic activity against three weed species *Melilotus officinalis* L. (Fabaceae), *Trigonella besseriana* Ser. (Fabaceae), and *Myosotis arvensis* (L.) Hill. (Boraginaceae) [27]. *M. officinalis* seed germination was inhibited by both juniper species; the germination of *M. arvensis* seeds was inhibited by *J. sabina* and stimulated by *J. excelsa* ; *T. besseriana* seeds were both inhibited and stimulated by both EO juniper species, depending on the doses. The root growth of these seeds was also affected. Some doses of the same oil are inhibitory, while others are stimulants. This generalized concept of “low-dose stimulation-high-dose inhibition” or “hormesis” was gradually supported by observations in the field. Exposure to new environments or to toxic substances increased the variability of phenotypic traits such as enzyme activity, morphological features, and growth. [28]. In a multi-species study [8], the antigerminative potential of 112 EOs from ninety-seven species belonging to sixteen families was evaluated. The authors found that α- and β-pinene, and limonene (Figure 6) were some of the most common constituents of the oils that suppress seed germination and seedling development; α-pinene and limonene were also present in the composition of the EO here reported.

Furthermore, another study [29] asserted that α-pinene was responsible for the antigerminative effect of *Ferula tunetana* Pomel ex Batt., essential oil on the seeds of *Medicago sativa*, *Triticum aestivum,* and *Lactuca sativa*. The phytotoxic activity of *Dracocephalum integrifolium* essential oil and, for the first time, of sabinene (Figure 7) on seeds of a dicotyledon *Amaranthus retroflexus* and of a monocotyledon *Poa annua* L., was reported [30]: the compound, at the highest doses used, exerted a significant inhibitory effect on the growth of *A. retroflexus* shoots compared to control. The antigerminative effect of sabinene was much more evident on shoot growth of *P. annua* seeds. The root growth of these seeds was also affected by the compound [30]. Since several essential oils with phytotoxic activities have sabinene as the main constituent, probably this compound could play an important role as an active compound of these oils.

According to recent studies, essential oils from *Drimys winterii* J.R. Forst. & G. Forst., *Nepeta flavida* Hub. Mor. and *Vitex agnus-castus* L., with a high percentage of sabinene, were evaluated for their possible phytotoxic effect against several weeds [31,32,33]. Moreover, De Martino and coworkers [34] reported the inhibitory effect of α-pinene and limonene, also present in *Juniperus horizontalis* EO, on radical elongation of *Raphanus sativus* L. and *Lepidium sativum* L. The herbicidal effect of EOs could also be related to 1,8-cineole, carvacrol, camphor, thymol, α-pinene, limonene, and volatile bioactive compounds with different levels of phytotoxicity [8,34,35]: these data are in agreement with a recent review [36], in which the data analysis reveals that terpenes, mainly mono- and sesquiterpenes, play a principal role in the phytotoxicity of EOs. Generally, an essential oil is a mixture of many substances in variable quantities. It is often not known if and how these can interact synergistically [16]. The examination of natural-based molecules to obtain eco-friendly herbicides requires specific target organisms and provides selective mode(s) of action. Some studies concerning the phytotoxic activity of EOs report their different impact on monocotyledonous and dicotyledonous plant species [37,38], depending on the constitution of the EO and the presence of specific dominant chemical groups.

### 3.3. Larvicidal Activity and Toxicity of J. horizontalis and of Sabinene

The results obtained in the larvicidal bioassay indicate the potential of juniper EO in limiting the survival of yellow mealworm. However, the larvicidal effect of *J. horizontalis* EO is several times lower in comparison to clove and cinnamon [39], garlic [40], or oregano essential oils [20], although it was not excluded that *J. horizontalis* EO or its main compound sabinene may exhibit some toxicity to non-target organisms, the doses that affected root growth were not toxic to either sludge worms or blood worms.

## 4. Materials and Methods

### 4.1. Volatile Oil

*J. horizontalis* (fresh leaves and branches) was collected in Wyoming, in the Bighorn Mountains. The identification was provided by Ms. Bonnie Heidel, a botanist at the Wyoming Natural Diversity Database, University of Wyoming. The specimens of the species were placed in the Herbarium of the University of Wyoming Rocky Mountains. The EO was obtained by steam water distillation, as described previously [4], with a yield of 1.3%. Pure EO (Voucher number EO_JH_2014 stored in Unipolab Presov, Slovakia) was brought to Slovakia by Prof. Zheljazkov for subsequent analysis.

### 4.2. GC-MS Analyses and Identification of Constituents

GC-MS analyses were conducted on a Varian 450-GC (Varian, Inc., Palo Alto, CA, USA) apparatus linked to a Varian 220-MS (Varian, Inc., Palo Alto, CA, USA) at the University of Presov in Slovakia. Component separation was achieved with a FactorFourTM capillary column VF 5 ms (30 m × 0.25 mm i.d., 0.25 μm film thickness). Injector type 1177 was warmed to a temperature of 220 °C. The injection way was without split (1 μL of a 1:1000 *n*-hexane solution). The carrier gas was Helium, with a flow rate of 1.2 mL/min. The column temperature was set with the following sequence: the initial temperature was 50 °C for 10 min, then increased to 100 °C at 3 °C/min, held isothermally for 5 min, and finally enhanced to 150 °C at 10 °C/min, for an overall analysis time of 46.67 min. The mass spectrometer trap was warmed to 200 °C, the manifold to 50 °C, and the transfer line to 270 °C. Mass spectra were scanned every 1 s in the range of 40–650 *m*/*z*. Most components were identified by comparing their Kovats retention indices (Ri) with those reported in the literature [41,42] or with those of standards available in our laboratory. The Kovats retention indices were calculated on the basis of a homologous series of n-alkanes (C_10_–C_35_) under the same operating conditions. Further identification of components was performed by comparing their mass spectra on both columns with either those present in NIST 02 and Wiley 275 libraries or with literature mass spectra [42,43,44] and in a personal library. The sample was analyzed in triplicate, and the standard deviation (SD) was calculated.

### 4.3. Herbicidal Activity

The possible herbicidal activity of the EO and of sabinene was tested on the seeds of *Hordeum vulgare* L. (barley), *Triticum aestivum* L. (wheat), *Sinapis alba* L. (white mustard), and *Lepidium sativum* L. (garden cress), generally accepted as crops and on seeds of *Trifolium pratense* L. (red clover), *Portulaca oleracea* L. (purslane), *Lolium perenne* L. (ryegrass) and *Barbarea vulgaris* L. (wintercress), generally accepted as weeds.

*Triticum aestivum* and *Hordeum vulgare* seeds were obtained from the Research center in Malý Šariš, Slovakia, in the year 2021; the seeds of *Trifolium pratense* var. *altaswede* and of *Sinapis alba* were obtained from company AgronaTeam Prešov, Slovakia; *Barbarea vulgaris*, *Portulaca oleracea, Lepidium sativum*, and *Lolium perenne* seeds were purchased from company FloraSeft, in chain store Hornabach Baumarkt AG (Bornheim, Rheinland-Pfalz, Germany).

The phytotoxicity was evaluated following the method previously reported [45] with some modifications. The EO and the pure compound solubilized in a water-acetone mixture (99.5:0.5) were tested at the doses of 0.5 μg/mL, 1.0 µg/mL, 5 µg/mL, 10 µg/mL, 50 µg/mL, and 100.0 μg/mL. No differences between controls performed with a water–acetone mixture and controls with water alone were detected. Five layers of Whatman filter paper and ten sterilized seeds were put in each Petri dish (Ø 90 mm). Finally, seven mL of each dose of EO solution or control was added to every Petri dish. Each treatment was repeated six times. The Petri dishes were placed in a growing chamber (Sanyo, MLR-351 H) with a natural photoperiod at 22 ± 1 °C. The number of germinated seeds and the root length (cm) were evaluated after 120 h.

### 4.4. Insecticidal Activity

The insecticidal (lethal) activity of *J. horizontalis* EO and of sabinene was assessed in a mortality test [20,40] using larvae of *Tenebrio molitor* L. (the yellow mealworm) (Coleoptera: Tenebrionidae). The mealworm beetle, *Tenebrio molitor* Linnaeus (Coleoptera: Tenebrionidae), is a pest of stored products, such as starches and pasta: this insect is able to infest broken grains of *Zea mays* (L.) and *T. aestivum* (Poales: Poaceae). Its presence in stored grain and bran may contaminate food with fragments of the body and feces by saprophytic microorganisms, producing a loss of food quality.

Individuals were obtained from their own breeding (Department of Ecology, FHaNS, University of Prešov, Slovakia). The EO (or sabinene) was dissolved in acetone; pure acetone was used as a control. Four doses of *J. horizontalis* EO and of sabinene were used in the larvicidal test corresponding to 435 µg Insect^−1^, 652.5 µg Insect^−1^, 870 µg Insect^−1^, and 1 740 µg Insect^−1^. One µL of solution of each dose of the samples was applied on the thorax of the larva of *T. molitor*, using a micropipette. Totally, 96 larvae were used for each dose. Bioassays were controlled after 24 h and 48 h, and the specimens were checked for movement and necrosis.

### 4.5. Toxicity

*Tubifex tubifex* Müller 1774 (sludge worm) (Annelida, Oligochaeta: Tubificidae)—standard model organism in ecotoxicological studies as well as *Chironomus aprillinus* Meigen 1830 (blood worms) (Diptera: Chironomidae) were used to test the toxicity of both *Juniperus horizontalis* EO and sabinene with the same doses used in herbicidal bioassay. Sludge worms were subjected to an express 3-min test for acute toxicity determination [46]. Because no mortality was observed after 3 min, prolonged exposition was applied: the number of death worms was checked after 10, 20, 30, 60, 180, and 240 min and, then, after 24 h. Blood worms were subjected to testing according to the standard methodology suggested by the World Health Organization (WHO, 2005) with slight modifications [47].

### 4.6. Statistics

The effect of the sample on germination was expressed as the average percentage of germinated seeds; the effect on the root growth was expressed as the average root length in cm determined using univariate statistics. Descriptive statistics were used to depict observed results. The overall differences between control and treatment, both in the germinative activity and in root growth, were assessed using *t*-TEST, with three levels of significance (*p* < 0.05; *p* < 0.01; *p* < 0.001). Spearman’s Rs correlation test was used to assess possible correlations between EO dose and germination activity, EO dose, and root length. Simple linear regression analysis was used to represent the observed significant correlations. All statistical analyses were performed using PAST version 2.17c [48]. The data from larvicidal bioassay on *Tenebrio molitor* were subjected to Finney’s probit analysis for determining the LD_50_, LD_90_, and 95% confidence intervals of upper/lower confidence limit (UCL/LCL) [49]. The crude mortality obtained from the larvicidal bioassay was corrected using Abbott’s formula (1925). Average larval mortality corrected from all replicates was determined using univariate statistics in the statistical program PAST [48].

## 5. Conclusions

The potential herbicidal effect of *J. horizontalis* EO was evaluated on different plant species. Although the results did not support our hypothesis, the effect of EO is evident. The weed group was less sensitive to applied EO than the crop group. The mode of action is not known: the cultivation process, developed to obtain the best quality of the requested crops (*Hordeum vulgare*, *Triticum aestivum*, *Sinapis alba,* and *Lepidium sativum*), could reduce the sensibility of these plants to the essential oil compounds. The species most commonly used for biological assays to test the potential herbicidal effect are selected from Poaceae and Brassicaceae. The root length of plant species belonging to Poaceae (*H. vulgare*, *T. aestivum*, *L. perenne*) and Brassicaceae (*S. alba*, *L. sativum*, and *B. vulgaris*) was inhibited by the EO of *J. horizontalis*. Suggestions for future study could be to focus not only on doses that have an influence on plants but also to identify more and less sensitive families or species. The results obtained in the larvicidal bioassay also indicate the potential of *J. horizontalis* EO in limiting yellow mealworm survival; the same doses used on seeds were not toxic to the sludge worm and blood worm, and these results can be a starting point for future research.

## Figures and Tables

**Figure 1 molecules-27-08408-f001:**
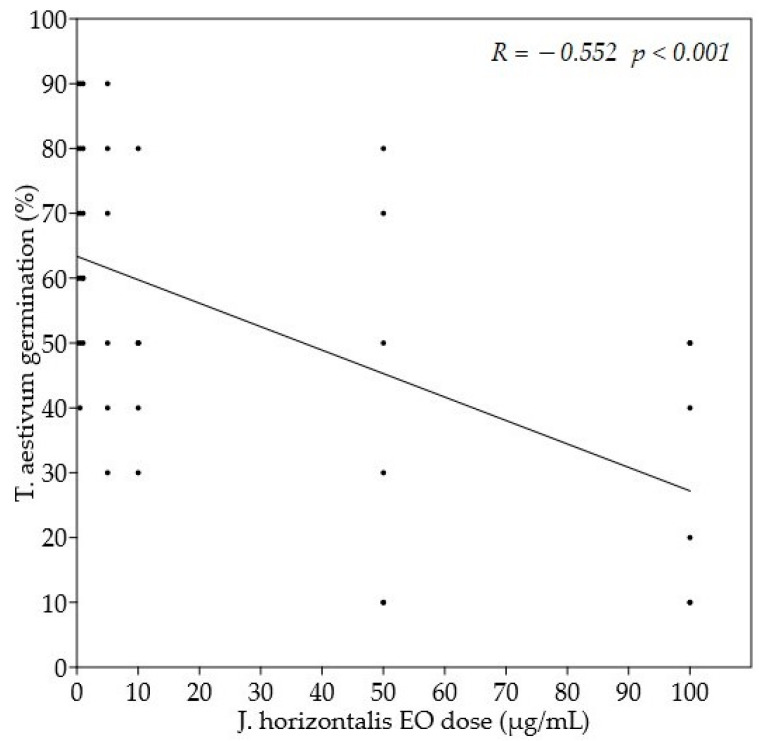
Simple linear regression between *J. horizontalis* EO dose (µg/mL) and *T. aestivum* germination (%).

**Figure 2 molecules-27-08408-f002:**
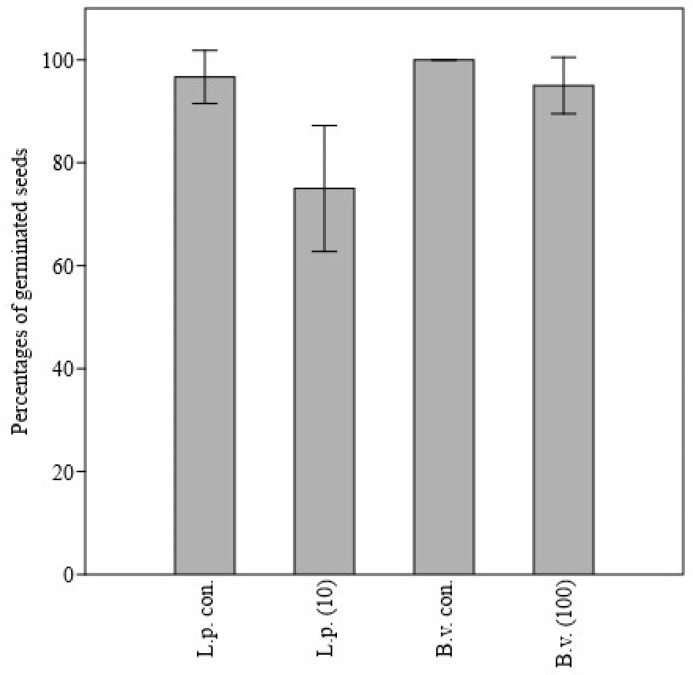
Bar-chart showing differences between *Lolium perenne* (L.p.) and *Barbarea vulgaris* (B.v.) percentages of germinated seeds in comparison to control (con.) after application of 10 µg/mL and 100 µg/mL of *Juniperus horizontalis* essential oil.

**Figure 3 molecules-27-08408-f003:**
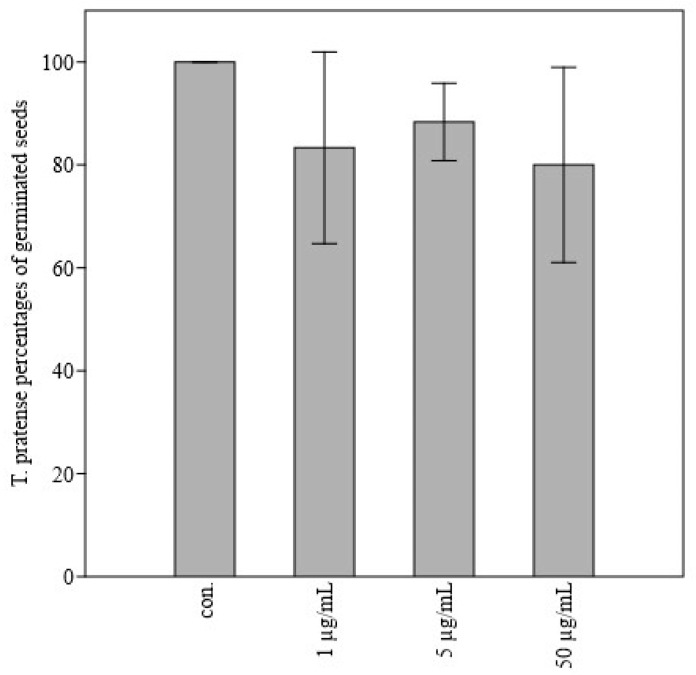
Bar-chart showing *Trifolium pratense* percentages of germinated seeds after application of 1 µg/mL, 5 µg/mL, and 50 µg/mL of *Juniperus horizontalis* essential oil in comparison to control.

**Figure 4 molecules-27-08408-f004:**
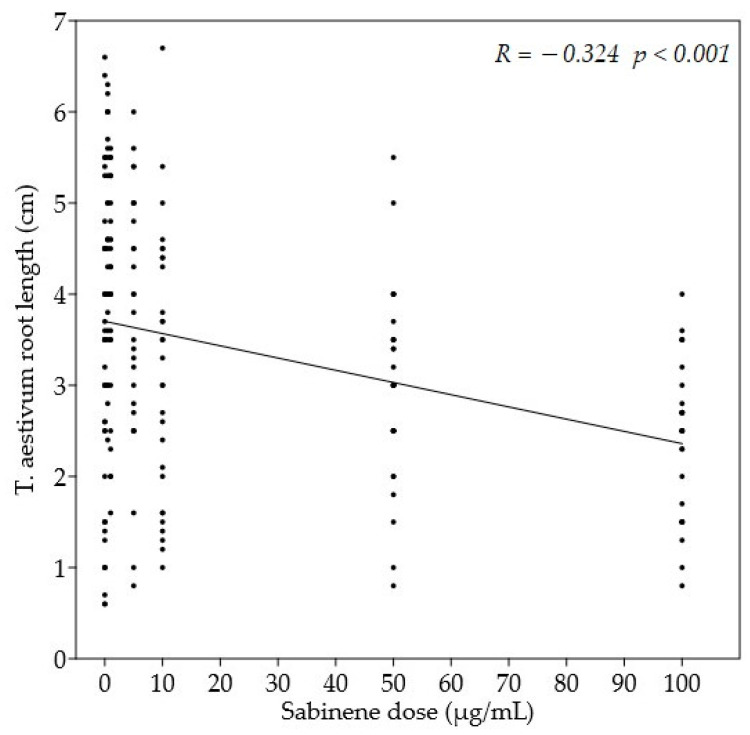
Simple linear regression between sabinene dose (µg/mL) and *T. aestivum* root length (cm).

**Figure 5 molecules-27-08408-f005:**
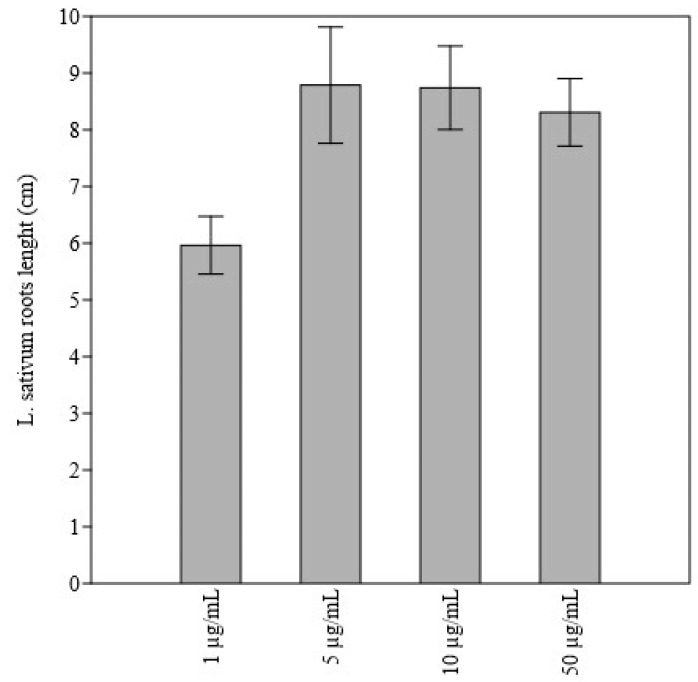
Bar-chart showing *L. sativum* roots length (cm) after application of 1 µg/mL in comparison to 5 µg/mL, 10 µg/mL, and 50 µg/mL doses of sabinene.

**Figure 6 molecules-27-08408-f006:**
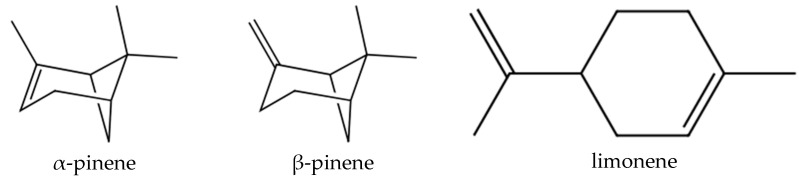
Some constituents of *Juniperus* oils.

**Figure 7 molecules-27-08408-f007:**
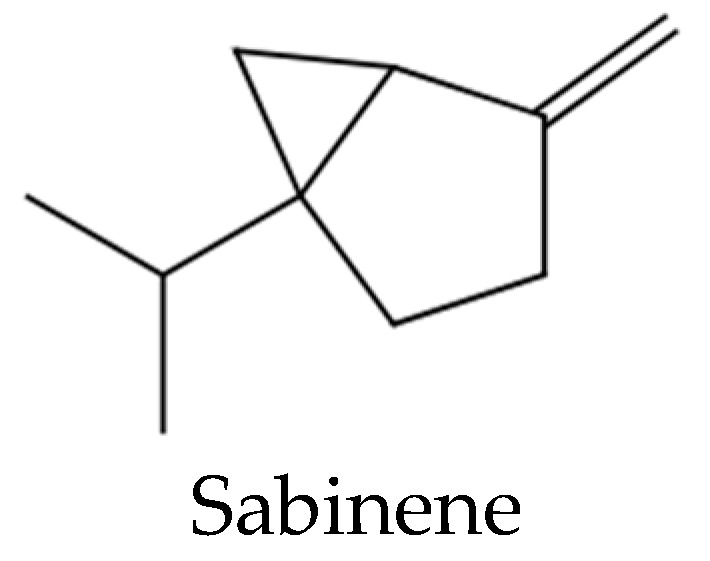
The main component of *Juniperus horizontalis* essential oil.

**Table 1 molecules-27-08408-t001:** Chemical composition of the essential oil of *Juniper horizontalis*.

Compound	% ± SD	RI ^a^	RI ^b^	Identification ^c^
α-Thujene	5.3 ± 0.2	932	931	KI, MS
α-Pinene	10.0 ± 0.9	936	939	Co-I, KI, MS
Sabinene	38.7 ± 2.0	973	976	KI, MS
β-Pinene	tr ^d^	978	978	Co-I, KI, MS
Myrcene	tr	987	991	Co-I, KI, MS
α-Terpinene	0.9 ± 0.2	1013	1018	Co-I, KI, MS
Limonene	7.8 ± 0.5	1025	1031	KI, MS
γ-Terpinene	8.3 ± 0.6	1051	1062	Co-I, KI, MS
*cis*-β-Terpineol	0.2 ± 0.0	1141	1144	KI, MS
1-Terpinen-4-ol	4.4 ± 0.1	1164	1163	KI, MS
α-Terpineol	0.3 ± 0.0	1176	1189	Co-I, KI, MS
Bornyl acetate	0.2 ± 0.0	1270	1285	Co-I, KI, MS
β-Elemene	0.8 ± 0.0	1389	1391	KI, MS
γ-Elemene	0.2 ± 0.0	1429	1433	KI, MS
β-Caryophyllene	0.6 ± 0.1	1467	1467	KI, MS
γ-Gurjunene	0.1 ± 0.0	1473	1473	KI, MS
γ-Muurolene	0.1 ± 0.0	1474	1477	KI, MS
Germacrene D	0.2 ± 0.0	1479	1480	KI, MS
*epi*-Bicyclosesquiphellandrene	0.1 ± 0.0	1487		KI, MS
Valencene	0.2 ± 0.0	1494	1491	KI, MS
α-Muurolene	0.3 ± 0.1	1496	1499	KI, MS
γ-Cadinene	0.7 ± 0.1	1507	1513	KI, MS
δ-Cadinene	1.2 ± 0.1	1520	1524	KI, MS
Elemol	8.6 ± 0.6	1541	1549	KI, MS
Spathulenol	0.2 ± 0.0	1572	1576	KI, MS
γ-Eudesmol	0.9 ± 0.1	1618	1630	KI, MS
Cubenol	0.6 ± 0.0	1630	1642	KI, MS
t-Cadinol	0.2 ± 0.0	1633	1642	KI, MS
t-Muurolol	2.3 ± 0.0	1633	1645	KI, MS
TOTAL	93.40			
Monoterpene hydrocarbons	71.00			
Oxygenated monoterpenes	5.1			
Sesquiterpene hydrocarbons	4.5			
Oxygenated sesquiterpenes	12.8			

^a^ = calculated retention index; ^b^ = literature retention index; ^c^ = Co-I = coinjection with authentic standard, RI = Retention index (comparison between software prediction and literature (Adams); MS = mass spectra); ^d^ tr-traces = <0.1%.

**Table 2 molecules-27-08408-t002:** Percentage (%) of germinated seeds after 120 h from the application of several doses of *J. horizontalis* EO. Results are reported as the mean ± SD of six experiments.

Model Plants		Mean Percentage of Germination by Influence of *J. horizontalis* EO						
		Doses (µg/mL)						
		100	50	10	5	1	0.5	Control
crops	*Sinapis alba* (Brassicaceae)	91.7 ± 4.1	93.3 ± 8.2	95.0 ± 5.5	96.7 ± 5.2	95.0 ± 5.5	98.3 ± 4.1	91.7 ± 13.3
	*Lepidium sativum* (Brassicaceae)	95.0 ± 12.2	98.3 ± 4.1	100.0 ± 0.0	98.3 ± 4.1	100.0 ± 0.0	100.0 ± 0.0	96.7 ± 5.2
	*Triticum aestivum* (Poaceae)	30.0 ± 19.90 **	41.7 ± 29.9	50.0 ± 16.7	60.0 ± 23.7	68.3 ± 14.7	65.0 ± 18.7	68.3 ± 14.7
	*Hordeum vulgare* (Poaceae)	91.7 ± 7.5	95.0 ± 5.5	93.3 ± 10.3	86.7 ± 13.7	93.3 ± 8.2	93.3 ± 8.2	90.0 ± 8.9
weeds	*Lolium perenne* (Poaceae)	80.0 ± 20.0	86.7 ± 15.0	75.0 ± 12.2 **	85.0 ± 13.8	91.7 ± 9.8	78.3 ± 21.3	96.7 ± 5.2
	*Portulaca oleracea* (Portulacaceae)	96.7 ± 5.2	98.3 ± 4.1	100.0 ± 0.0	96.7 ± 5.2	100.0 ± 0.0	98.3 ± 4.1	100.0 ± 0.0
	*Barbarea vulgaris* (Brassicaceae)	95.0 ± 5.5 *	96.7 ± 5.2	96.7 ± 8.2	96.7 ± 5.2	98.3 ± 4.08	100.0 ± 0.0	100.0 ± 0.0
	*Trifolium pratense* (Fabaceae)	91.7 ± 11.7	80.0 ± 19.0	86.7 ± 19.7	88.3 ± 7.5 **	83.3 ± 18.6 *	93.3 ± 8.7	100.0 ± 0.0

Statistically significant differences are marked by the stars (*); * *p* ≤ 0.05; ** *p* ≤ 0.01 vs. control, according to *t*-TEST with three levels of significance.

**Table 3 molecules-27-08408-t003:** Root length (cm) after 120 h from application of several doses of *J. horizontalis* EO. Results are reported as the mean ± SD of six experiments.

Model Plants		Root Lenght by Influence of *J. horizontalis* EO [cm]						
		Doses [µg/mL]						
		100	50	10	5	1	0.5	Control
crops	*Sinapis alba* (Brassicaceae)	1.5 ± 0.5 ***	2.2 ± 0.7 ***	2.5 ± 1.2 *	1.8 ± 0.9 **	2.6 ± 0.5 *	2.5 ± 0.9 *	4.5 ± 1.2
	*Lepidium sativum* (Brassicaceae)	2.4 ± 0.5 ***	4.4 ± 2.5	3.8 ± 0.9 **	2.8 ± 0.6 ***	3.9 ± 1.8 *	3.7 ± 1.7 **	6.1 ± 1.1
	*Triticum aestivum* (Poaceae)	0.8 ± 0.4 **	1.6 ± 1.0 **	1.0 ± 0.4 **	1.5 ± 0.6 **	1.4 ± 0.6 **	1.4 ± 0.4 **	3.2 ± 0.7
	*Hordeum vulgare* (Poaceae)	2.7 ± 0.3 **	3.1 ± 0.4 **	3.2 ± 0.3 **	3.1 ± 0.6 **	3.0 ± 0.6 **	3.4 ± 0.3 **	4.3 ± 0.4
weeds	*Lolium perenne* (Poaceae)	1.4 ± 0.2 **	1.8 ± 0.2 **	1.7 ± 0.3 **	1.9 ± 0.3 **	1.8 ± 0.5 **	1.7 ± 0.5 **	2.5 ± 0.2
	*Portulaca oleracea* (Portulacaceae)	1.8 ± 0.2 *	2.2 ± 0.3 **	2.1 ± 0.1 ***	1.8 ± 0.3 *	1.0 ± 0.1 ***	1.6 ± 0.2	1.5 ± 0.2
	*Barbarea vulgaris* (Brassicaceae)	0.8 ± 0.1 *	1.0 ± 0.1	1.1 ± 0.1	0.9 ± 0.2	0.8 ± 0.2 *	0.9 ± 0.1	1.2 ± 0.4
	*Trifolium pratense* (Fabaceae)	1.7±	2.9 ± *	1.9±	2.3 ± **	1.7±	2.1 ± *	1.3±

Statistically significant differences are marked by the stars (*); * *p* ≤ 0.05; ** *p* ≤ 0.01; *** *p* ≤ 0.001 vs. cont, according to *t*-TEST with three levels of significance.

**Table 4 molecules-27-08408-t004:** Percentage (%) of germinated seeds after 120 h from the application of several doses of sabinene. Results are reported as the mean ± SD of six experiments.

Model Plants		Mean Percentage of Germination by Influence of Sabinene [%]						
		Doses [µg/mL]						
		100	50	10	5	1	0.5	Control
crops	*Sinapis alba* (Brassicaceae)	83.3 ± 5.8	96.7 ± 5.8	93.3 ± 11.5	100.0 ± 0.0	93.3 ± 5.8	86.7 ± 15.3	91.7 ± 13.3
	*Lepidium sativum* (Brassicaceae)	96.7 ± 5.8	100.0 ± 0.00	100.0 ± 0.0	100.0 ± 0.0	100.0 ± 0.0	100.0 ± 0.0	100.0 ± 0.0
	*Triticum aestivum* (Poaceae)	90.0 ± 17.3	93.3 ± 5.8 *	96.7 ± 5.8 *	90.0 ± 0.0 *	80.0 ± 17.3	83.3 ± 11.6	68.3 ± 14.7
	*Hordeum vulgare* (Poaceae)	86.7 ± 15.3 *	100.0 ± 0.0	96.7 ± 5.8	93.3 ± 5.8 *	96.7 ± 5.8	96.7 ± 5.8	100.0 ± 0.0
weeds	*Lolium perenne* (Poaceae)	70.0 ± 10.0 ***	96.7 ± 5.8	90.0 ± 17.3	73.3 ± 25.2 *	90.0 ± 0.0	86.7 ± 5.8 *	96.7 ± 5.2
	*Portulaca oleracea* (Portulacaceae)	100.0 ± 0.0	93.3 ± 5.8	96.6 ± 5.8	96.7 ± 5.8	96.7 ± 5.8	100.0 ± 0.0	96.7 ± 5.2
	*Barbarea vulgaris* (Brassicaceae)	93.3 ± 5.8	100.0 ± 0.0	93.3 ± 5.8	100.0 ± 0.0	96.7 ± 5.8	100.0 ± 0.0	96.7 ± 5.2
	*Trifolium pratense* (Fabaceae)	73.3 ± 11.5	86.7 ± 15.3	76.7 ± 20.8	70.0 ± 20.0	90.0 ± 17.3	56.7 ± 23.1	80.0 ± 20.0

Statistically significant differences are marked by the stars (*); * *p* ≤ 0.05; *** *p* ≤ 0.001 vs. control, according to *t*-TEST with three levels of significance.

**Table 5 molecules-27-08408-t005:** Root length (cm) after 120 h from the application of several doses of sabinene EO. Results are reported as the mean ± SD of six experiments.

Model Plants		Root Lenght by Influence of Sabinene [cm]						
		Doses [µg/mL]						
		100	50	10	5	1	0.5	Cont.
crops	*Sinapis alba* (Brassicaceae)	2.3 ± 0.8	4.5 ± 1.6	4.8 ± 1.6	4.2 ± 0.6	3.5 ± 1.0	2.8 ± 1.8	4.5 ± 1.2
	*Lepidium sativum* (Brassicaceae)	5.8 ± 1.7	8.3 ± 0.6 **	8.7 ± 0.7 **	8.8 ± 0.6 **	6.0 ± 1.0	7.0 ± 0.5	6.1 ± 1.1
	*Triticum aestivum* (Poaceae)	2.5 ± 0.4	3.0 ± 0.4	3.2 ± 0.2	3.7 ± 0.3	4.0 ± 0.5	4.6 ± 0.2 *	3.2 ± 0.7
	*Hordeum vulgare* (Poaceae)	3.7 ± 1.5	4.3 ± 1.2	4.5 ± 1.9	3.7 ± 0.5	4.3 ± 0.5	9.0 ± 6.0	3.9 ± 0.7
weeds	*Lolium perenne* (Poaceae)	1.3 ± 0.2 ***	1.6 ± 0.2 ***	2.0 ± 0.3 *	2.0 ± 0.3 *	1.8 ± 0.5 *	2.3 ± 0.2	2.5 ± 0.2
	*Portulaca oleracea* (Portulacaceae)	1.8 ± 0.3	1.9 ± 0.0 **	2.0 ± 0.1 **	1.9 ± 0.3 *	2.1 ± 0.2 ***	2.2 ± 0.1 ***	1.5 ± 0.1
	*Barbarea vulgaris* (Brassicaceae)	0.8 ± 0.1	0.8 ± 0.1	1.1 ± 0.2	1.0 ± 0.0	1.0 ± 0.2	1.3 ± 0.2	1.3 ± 0.4
	*Trifolium pratense* (Fabaceae)	1.9 ± 0.0	2.2 ± 0.2 *	2.2 ± 0.9	1.6 ± 0.3	2.0 ± 0.2	2.0 ± 1.1	1.6 ± 0.4

Statistically significant differences are marked by the stars (*); * *p* ≤ 0.05; ** *p* ≤ 0.01; *** *p* ≤ 0.001 vs. control, according to *t*-TEST with three levels of significance.

**Table 6 molecules-27-08408-t006:** Average mortality (in % ± standard deviation) of yellow mealworm *Tenebrio molitor* larvae after 48 h exposure to juniper EO.

EO Dose in µg Insect^−1^	Average Mortality in % ± SD
1740	93.70 ± 5.94%
870	57.98 ± 11.88%
652.5	14.07 ± 10.44%
435	12.82 ± 6.25%

## Data Availability

Not applicable.

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
