# Peer review of "Juniperus horizontalis Moench: Chemical Composition, Herbicidal and Insecticidal Activities of Its Essential Oil and of Its Main Component, Sabinene"

_molecules, 2022, doi:10.3390/molecules27238408_

Round 1

Reviewer 1 Report

The reviewed manuscript contains the results of a few laboratory experiments assessing the effect of the essential oil of Juniperus horizontalis and sabinene on plant germination and insect activity.

The general comments:

1.       The manuscript chapters are not well written.

2.       The abstract does not correspond with the findings! It is also chaotic. Passages about phytotoxic studies are mixed with those with insecticidal essays. Organize that.

3.       Why do you correlate doses in phytotoxic studies with those for insecticidal studies? Do the studied insects coexist with weeds?

4.       Abstract – give the final time of assessments for herbicidal and insecticidal tests.

5.       Ln 17 – How are the "toxic" activities different from herbicidal and larvicidal?

6.       Please, explain in the abstract that 3 crop species and 3 weed species were tested.

7.       The concluding sentence is missing in the abstract. Does this oil exert such great phytotoxic and insecticidal effects? Based on the results, I am not convinced.

8.       Keywords: delete those keywords that are repeated in your title. Should all the insects be mentioned

9.       The Introduction is chaotic and does not introduce the reader to the aims of this work in a coherent way. Some parts are too long, e.g., lines 60-80.

10.   Introduction: the authors introduce the term "allelopathy," but later in the manuscript, this expression is not utilized effectively, so what was a reason to explain it?

11.   Introduction: ln 41 – what does "shrubby" mean?

12.   Introduction: could you elaborate more on the chemical composition of. J. horizontalis in that part? Especially on sabinene.

13.   Lines 48-50 do not correlate with the paragraph. Also, in another paragraph, the authors mention phytotoxicity again. That creates chaos.

14.   ln 46 "as a" instead of "to cover."

15.   ln 53 – what does the word "phenomena" refer to?

16.   Ln 54 – the term "allelochemicals" is introduced for the first time; many readers may not be familiar with this specialist term.

17.   Ln 55, what do you mean by "place of production"? Plant organ?

18.   Ln 78 – what is "undesider"?

19.   Ln 88, please give a citation

20.   Why did you study those insects? That is not clear. Especially blood worms?

21.   Ln 99 and the title – "biological activity" or "biological effect" has a broad meaning. I suggest narrowing the meaning to herbicidal and insecticidal.

22.   Ln 96 – why "could have"? The research hypothesis should be unambiguous.

23.   Ln 90: delete as. It changes the meaning of the sentence completely.

24.   Ln 89 and 95 – delete "So" at the beginning of the sentences.

25.   ln 90 – be strict in using phrases: phytotoxic effect on germination …

26.   ln 101 – use Italic for the Latin name of species

27.   Your resulting composition of the EO is mainly of monoterpene hydrocarbons. Based on your Discussion: why did you expect herbicidal and insecticidal effects if they are related to other oxygenated terpenes? Sabinene is also in the group of monoterpene hydrocarbons.

28.   What was the yield of Juniperus oil?

29.   Ln 114: Table 2 reports

30.   Table 2 is not self-explanatory. What are the units of concentration? What was the time of assessment? In days/hours? What was the number of replications? What is the SD?

31.   Statistics: if you compare each concentration with the control only, one-way ANOVA is not justified. I sugest using t-test. Also, did you check the assumptions of ANOVA? Working with percentage values means you should perform the transformation of numeric data if they are in a narrow range like 90-100% or 60-30%, etc.

32.   Moreover, based on Table 2, the germination of wheat was poor even for control. How old were the seeds? Maybe that is a reason for their poor outcome, and not necessarily the phytotoxicity of the oil.

33.   Again, do you have any justification for the doses you used? They are quite low, especially considering the oil's dominating group of sesquiterpene hydrocarbons. That could be the reason for the poor effect of the oil on seed germination.

34.   You don't give any SD or SE in the table; that is why the significance of ** and * by the same number look strange for Trifolium.

35.   Figure 1, please delete it. It repeats the data shown in Table2.

36.   Figure 2 – what is n=? Also, did you perform this analysis on the mean or raw values? A higher number of input values is needed to turn a valuable outcome of correlation.

37.   The title of chapter 2.3 is confusing. Did you measure root growth or root length? That is not the same! Be precise.

38.   Ln 158 – did you calculate the "germination rate"? Please be cautious with the terms.

39.   Table 3 is also not informative: time of assessment, SD, number of replications, etc.

40.   Ln 164 – Table 4 reports.

41.   Ln 178 – again, root growth or length

42.   ln 180 – root or radical?

43.   Tables 4 and 5– the same comments as the previous tables.

44.   Also, for all analyses presented in tables – I suggest performing a t-test!

45.   Figure 3 – delete.

46.   Figure 4 – the same comments as for Figure 2.

47.   ln 224 – LC or LD? Check M&M

48.   For insects, you use Latin names and then common names. Be consequent. Not all of the readers are familiar with both.

49.   Present the numeric values for chapter 2.6 in the table to make it easier to follow.

50.   Ln 229 – was this "slight mortality" significant? If not, do not report that at all.

51.   Discussion, please give the origin of the EO, as M&M is described later. It is important to help readers to follow the presented results.

52.   The Discussion is too long. The oil did not affect seed germination in a significant range, so it is better to combine those results with the effects on root length and discuss it jointly.

53.   A huge part of the Discussion does not relate to Juniperus oil. Some speculations about phytotoxicity are r too broad, i.e., why do you compare different oils of different compositions tested on different weeds? Is there any relation between your and other studies?

54.   Moreover, whenever you discuss the toxicity effect of other authors, give the doses they study to help the reader. Giving the EO doses would clarify the outcomes.

55.   Ln 272 – what do you mean by the expression "The causes at the basis of such increases"? It is not clear.

56.   ln 274 – delete "studied."

57.   ln 286 – delete "simply poa."

58.   Lines 305-316 – move to the Introduction

59.   Lines 317-339 – delete, as this part is irrelevant to your results.

60.   Lines 341-343 – why do you compare Juniperus oil with the cited oils? They have different groups of main compounds. Find oils of similar composition of main chemical groups for comparisons.

61.   Chapter 3.5 is unnecessary – this part is not well discussed and could be combined with the previous chapter.

62.   Ln 382 – seeds of crops and weeds. Give varieties of crops you used. How old were they? Especially wheat?

63.   Ln 391 – Did you perform 6 experiments or have 6 repetitions?

64.   Ln 393 – assessment time must be given in all tables and figures.

65.   Statistics – change according to my suggestions to make it fitted to your specific results.

66.   I can't entirely agree with your Conclusions, lines 443-449. The effects are not so clear. The oil inhibits root length but does not change the number of germinated seeds.

67.   Ln. 449 – you did not study root elongation but root length.

68.   I don't find your speculation about crops being less sensitive as merit. Why would crops be more resistant to allelochemicals? And – why do you mention in this sentence allelochemicals again? That is a broad term. Be precise – essential oil compounds.

69.   English language, syntax, and grammar must be corrected in all the text.

Author Response

The reviewed manuscript contains the results of a few laboratory experiments assessing the effect of the essential oil of Juniperus horizontalis and sabinene on plant germination and insect activity.

The general comments:

  1. The manuscript chapters are not well written.

Answer: The manuscript chapters were written in a better way.

  1. The abstract does not correspond with the findings! It is also chaotic. Passages about phytotoxic studies are mixed with those with insecticidal essays. Organize that.

Answer: The abstract was organized in a better way.

  1. Why do you correlate doses in phytotoxic studies with those for insecticidal studies? Do the studied insects coexist with weeds?

Answer: Phytotoxic studies were not corelated with those insecticidal. Insecticidal studies were performed with the yellow mealworm (Tenebrio molitor), employing the different (effective) doses of EO and/or sabinene in comparison to phytotoxic studies (please, see Material and methods). Probably, the question is targeted on the toxicity studies with Tubifex tubifex and Chironomus aprilinus – both model organisms are used in the studies to prove non-toxicity/low negative effect of applied essential oils. T. tubifex is in general taken as a representative of soil anelids and Chi. aprilinus is in general taken as a representative of water invertebrates. Because we wanted to prove, that just the doses which impacted root growth are simultaneously not toxic to sludge worms or blood worms, the identical doses of herbicidal assay were applied. It is standard procedure in toxicity assay, to test the doses, which cause the mortality/phytotoxicity, simultaneously on the non-target model organisms. That is why we did it in same way.

  1. Abstract – give the final time of assessments for herbicidal and insecticidal tests.

Answer: The final time of herbicidal activity was 120 h. The final times for insecticidal test were 24 and 48 hours. The information were also added in the abstract

  1. Ln 17 – How are the "toxic" activities different from herbicidal and larvicidal?

Answer: As mentioned above, the toxicity test is standardly applied, when the non-negative effect/low toxicity of applied substance want to be proven – it would be the best ,combination, i.e. when substance would be phytotoxically active but, in same time would not harm the organisms, which are non-target . It is standard to prove, that effective dose which in our case caused some phytotoxic effect would be on the other side safe for non-target organisms and would not cause environmental damage after possible application in water or soil.

  1. Please, explain in the abstract that 3 crop species and 3 weed species were tested.

Answer: The requested information was added in the text

  1. The concluding sentence is missing in the abstract. Does this oil exert such great phytotoxic and insecticidal effects? Based on the results, I am not convinced.
    Answer: The concluding sentence was added in the text to clarify our results that could be the starting point for future researches.

  1. Keywords: delete those keywords that are repeated in your title. Should all the insects be mentioned

Answer: The keywords were modified as requested

  1. The Introduction is chaotic and does not introduce the reader to the aims of this work in a coherent way. Some parts are too long, e.g., lines 60-80.

Answer: The sentences were rewritten and shorten

  1. Introduction: the authors introduce the term "allelopathy," but later in the manuscript, this expression is not utilized effectively, so what was a reason to explain it?

Answer: The term allelopathy was removed as suggested by referee and some sentences were changed and /or deleted.

  1. Introduction: ln 41 – what does "shrubby" mean?

Answer: The term shrubby was corrected with shrub

  1. Introduction: could you elaborate more on the chemical composition of. J. horizontalis in that part? Especially on sabinene.

Answer: Sabinene was introduced also in the Introduction paragraph, as suggested

  1. Lines 48-50 do not correlate with the paragraph. Also, in another paragraph, the authors mention phytotoxicity again. That creates chaos.

Answer: The sentences were rewritten.

  1. ln 46 "as a" instead of "to cover."

Answer: The term was corrected as requested

  1. ln 53 – what does the word "phenomena" refer to?

Answer: the term was deleted

  1. Ln 54 – the term "allelochemicals" is introduced for the first time; many readers may not be familiar with this specialist term.

Answer: The term allelochemicals was substituted, as suggested.

  1. Ln 55, what do you mean by "place of production"? Plant organ?

Answer: the term place was changed in organ as suggested.

  1. Ln 78 – what is "undesider"?

Answer: the term undesider was substituted with unwanted

  1. Ln 88, please give a citation
    Answer: A citation was added: Plata-Rueda, A.; Zanuncio, J.C.; Serrão, J.E.; Martínez, L.C. Origanum vulgare Essential Oil against Tenebrio molitor (Coleoptera: Tenebrionidae): Composition, Insecticidal Activity, and Behavioral Response. Plants 2021, 10, 2513
  2. Why did you study those insects? That is not clear. Especially blood worms?

As indicated in MS, yellow mealworms represent pests of stored products – testing of essential oil can thus indicate the EO practical usage in the stored products protection against mealworms. As mentioned above, T. tubifex is in general taken as a representative of soil anelids and Chi. aprilinus is in general taken as a representative of water invertebrates. Expect, f.e. Daphnia can be also used.

  1. Ln 99 and the title – "biological activity" or "biological effect" has a broad meaning. I suggest narrowing the meaning to herbicidal and insecticidal.

Answer:  The term “biological activity” was changed, as suggested.

  1. Ln 96 – why "could have"? The research hypothesis should be unambiguous.

Answer: The term “could” was deleted, as suggested.

  1. Ln 90: delete as. It changes the meaning of the sentence completely.

Answer: The term “as” was deleted, as suggested.

  1. Ln 89 and 95 – delete "So" at the beginning of the sentences.

Answer: The term “so” was deleted, as suggested.

  1. ln 90 – be strict in using phrases: phytotoxic effect on germination …

Answer: The sentence was shortened, as requested.

  1. ln 101 – use Italic for the Latin name of species

Answer: Latin name was used, as requested.

  1. Your resulting composition of the EO is mainly of monoterpene hydrocarbons. Based on your Discussion: why did you expect herbicidal and insecticidal effects if they are related to other oxygenated terpenes? Sabinene is also in the group of monoterpene hydrocarbons.

Answer: As reported in the text, the herbicidal effect of EOs could be also related to 1,8-cineole, carvacrol, but also to α-pinene, limonene and others, that are monoterpene hydrocarbons: the possible synergistic effects of the volatile compounds in the essential oil was also reported in literature

  1. What was the yield of Juniperus oil?

Answer: The yield was added.

  1. Ln 114: Table 2 reports

Answer: The term was corrected as suggested.

  1. Table 2 is not self-explanatory. What are the units of concentration? What was the time of assessment? In days/hours? What was the number of replications? What is the SD?

Answer: The number of replications, the units of concentrations, the time of assessment were inserted in the table, as suggested.

  1. Statistics: if you compare each concentration with the control only, one-way ANOVA is not justified. I suggest using t-test. Also, did you check the assumptions of ANOVA? Working with percentage values means you should perform the transformation of numeric data if they are in a narrow range like 90-100% or 60-30%, etc.

Answer: T test was performed and reported in the paper, as suggested

  1. Moreover, based on Table 2, the germination of wheat was poor even for control. How old were the seeds? Maybe that is a reason for their poor outcome, and not necessarily the phytotoxicity of the oil.

Answer: Although the germination of the control seeds was 68.3%, the germination percentage of the treated seeds was more than 30% lower (30% at dose 100 µg/ml)

  1. Again, do you have any justification for the doses you used? They are quite low, especially considering the oil's dominating group of sesquiterpene hydrocarbons. That could be the reason for the poor effect of the oil on seed germination.

Answer: The doses used were established on the basis of literature data. Following, a paper in which almost the same doses (from 100 to 0.0625 µg/ml) were employed:

Gruľová, D.; Caputo, L.; Elshafie, H.S.; Baranová, B.; De Martino, L.; Sedlák, V.; Gogaľová, Z.; Poráčová, J.; Camele, I.; De Feo, V. Thymol Chemotype Origanum vulgare L. Essential Oil as a Potential Selective Bio-Based Herbicide on Monocot Plant Species. Molecules 202025, 595. https://doi.org/10.3390/molecules25030595

  1. You don't give any SD or SE in the table; that is why the significance of ** and * by the same number look strange for Trifolium.

Answer: SD was inserted in each table, as requested.

  1. Figure 1, please delete it. It repeats the data shown in Table2.

Answer: Figure 1 was deleted as suggested.

  1. Figure 2 – what is n=? Also, did you perform this analysis on the mean or raw values? A higher number of input values is needed to turn a valuable outcome of correlation.
    Answer: The analyses performed were based on the mean values (n = 6), but, according to your suggestion the figure 2 was corrected with the raw values.

  1. The title of chapter 2.3 is confusing. Did you measure root growth or root length? That is not the same! Be precise.

Answer: The title was changed, as requested

  1. Ln 158 – did you calculate the "germination rate"? Please be cautious with the terms.

Answer: The term “rate” was deleted.

  1. Table 3 is also not informative: time of assessment, SD, number of replications, etc.

Answer: The number of replications, the units of concentrations, the time of assessment were inserted in the table, as suggested.

  1. Ln 164 – Table 4 reports.

Answer: The term was corrected as suggested.

  1. Ln 178 – again, root growth or length

Answer: The title was also changed, as requested

  1. ln 180 – root or radical?

Answer: The term radical was corrected.

  1. Tables 4 and 5– the same comments as the previous tables.

Answer: The number of replications, the units of concentrations, the time of assessment were inserted in the tables, as suggested

  1. Also, for all analyses presented in tables – I suggest performing a t-test!

Answer: T test was performed and reported for all analyses.

  1. Figure 3 – delete.

Answer: Figure 3 was deleted, as suggested.

  1. Figure 4 – the same comments as for Figure 2.
    Answer: The analyses performed were based on the mean values (n = 6), but, according to your suggestion the figure 2 was corrected with the raw values.
  2. ln 224 – LC or LD? Check M&M
    Answer: The term was LD and was corrected
  3. For insects, you use Latin names and then common names. Be consequent. Not all of the readers are familiar with both.
    Answer: The terms were corrected.
  4. Present the numeric values for chapter 2.6 in the table to make it easier to follow.
    Answer: The table was added, as suggested
  5. Ln 229 – was this "slight mortality" significant? If not, do not report that at all.
    Answer: The sentence was deleted, as suggested.
  6. Discussion, please give the origin of the EO, as M&M is described later. It is important to help readers to follow the presented results.

Answer: The information about the origin of the oil was inserted also in the discussion, also suggested.

  1. The Discussion is too long. The oil did not affect seed germination in a significant range, so it is better to combine those results with the effects on root length and discuss it jointly.

Answer: The results were combined as suggested; also results of sabinene were combined in the same way. The two paragraphs of discussion were also combined. Moreover, the discussion was shortened, as requested.

  1. A huge part of the Discussion does not relate to Juniperus oil. Some speculations about phytotoxicity are too broad, i.e., why do you compare different oils of different compositions tested on different weeds? Is there any relation between your and other studies?

Answer: The discussion was based on similar studies with similar components to make a comparison

  1. Moreover, whenever you discuss the toxicity effect of other authors, give the doses they study to help the reader. Giving the EO doses would clarify the outcomes.

Answer: in the cited paper, the doses of the tested oil was reported as follow:

  • The OEO was prepared in 2.5 mL of acetone to obtain six dilutions (1.56, 3.12, 6.25, 12.5, 25, and 50 µg.insect−1) (Reference number 20).
  • Six concentrations of essential oil besides the control (acetone) were adjusted in 1mL of stock solution (essential oil and acetone): 1, 2, 4, 8, 16 and 32% (w/v) (Reference number 40).

The aim was to compare the results and the principles of the study.

  1. Ln 272 – what do you mean by the expression "The causes at the basis of such increases"? It is not clear.

Answer: the sentence was deleted.

  1. ln 274 – delete "studied."

Answer: the term “studied” was deleted as requested.

  1. ln 286 – delete "simply poa."

Answer: the term “simply poa” was deleted as requested.

  1. Lines 305-316 – move to the Introduction

Answer: Lines 305-316 were deleted, as suggested.

  1. Lines 317-339 – delete, as this part is irrelevant to your results.

Answer: Lines 317-339 were deleted, as suggested.

  1. Lines 341-343 – why do you compare Juniperus oil with the cited oils? They have different groups of main compounds.

Answer: Because the Authors also used yellow mealworms Tenebrio molitor as model organisms to prove insecticidal activity. Till the time of manuscript submission, we were honestly not able to find any paper evaluating insecticidal activity of pure sabinene or an EO with sabinene as one of main compounds against mealworms. 

  1. Chapter 3.5 is unnecessary – this part is not well discussed and could be combined with the previous chapter.
    Answer: the part was combined, as suggested.
  2. Ln 382 – seeds of crops and weeds. Give varieties of crops you used. How old were they? Especially wheat?

Answer: The requested information were added in M&M

  1. Ln 391 – Did you perfom 6 experiments or have 6 repetitions?

Answer: Six experiments were conducted.

  1. Ln 393 – assessment time must be given in all tables and figures.

Answer: The assessment time was added.

  1. Statistics – change according to my suggestions to make it fitted to your specific results.
    Answer: Statistics was changed, as suggested.
  2. I can't entirely agree with your Conclusions, lines 443-449. The effects are not so clear. The oil inhibits root length but does not change the number of germinated seeds.

Answer: The conclusion was modified as requested, without reference to effects on germination.

  1. Ln. 449 – you did not study root elongation but root length.

Answer: The term elongation was corrected with the term length

  1. I don't find your speculation about crops being less sensitive as merit. Why would crops be more resistant to allelochemicals? And – why do you mention in this sentence allelochemicals again? That is a broad term. Be precise – essential oil compounds.

Answer: The conclusion was modified and the term “allelochemicals” was substituted with essential oil compounds, as requested.

  1. English language, syntax, and grammar must be corrected in all the text.

Answer: English language, syntax, and grammar was corrected by the native speaker and University English teacher Dr. Jonathan Gresty.

Reviewer 2 Report

Dear authors

The manuscript is important in representing allelopathy study on coniferous tree which is intersiting and relevant.

Some general comments, some abbreviations were not disclosed. Species names need to be italicised, checked throughout the text. All symbol need to be italicised, checked throughout the text.

Ln 89-92, please disclosed the reason why specific crops and weeds were used. Why mealworm, sludge worm and blood worm were used? Are there any correlation with real pest etc?

Ln 108, what is RI? Pleased add it under the table.

Ln 133, please present the data as bar chart

Ln 138, combine together with other species

Ln. 159, how many replications were conducted for this experiment? As there is no SD

Ln 202, please present the data as bar chart as it will help to see the trend

Ln 260, section 3.2

Please add molecular structure while discussing their biological properties

Ln 292, section 3.3

Please add molecular structure while discussing their biological properties

References need to be adjusted according to the journal guideline.

Author Response

The manuscript is important in representing allelopathy study on coniferous tree which is intersiting and relevant.

Some general comments, some abbreviations were not disclosed. Species names need to be italicised, checked throughout the text. All symbol need to be italicised, checked throughout the text.

  1. Ln 89-92, please disclosed the reason why specific crops and weeds were used.

Answer: In the current research were used plant species as a crops and weeds selected according availability in our locality. The selection was made based on the other publications where were used the same species as crops or weeds.

  1. Why mealworm, sludge worm and blood worm were used? Are there any correlation with real pest etc?

Answer: Mealworm were used to prove insecticidal activity of juniperus EO and its main compound because the mealworm represent one of the pest of stored products, such as starches and pasta: this insect is able to infest broken grains of Zea mays (L.) and Triticum aestivum L. Its presence in stored grain and bran may contaminate food with fragments of the body, faeces and by saprophytic microorganisms producing loss of food quality - as indicated in the manuscript. Tubifex tubifex and Chironomus aprilinus – both model organisms are used in the studies in aim to prove non-toxicity/low negative effect of applied essential oils. T. tubifex is in general taken as a representative of soil anelids and Chi. aprilinus is in general taken as a representative of water invertebrates.

  1. Ln 108, what is RI? Pleased add it under the table.

Answer: RI = retention index – it was corrected under the table

  1. Ln 133, please present the data as bar chart

           Answer: Bar chart was added.

  1. Ln 138, combine together with other species

Answer: Done.

  1. 159, how many replications were conducted for this experiment? As there is no SD

Answer: Each treatment was repeated for six times, as mentioned in M&M. SD was also added.

  1. Ln 202, please present the data as bar chart as it will help to see the trend

Answer: Bar-chart was added.

  1. Ln 260, section 3.2 Please add molecular structure while discussing their biological properties

Answer: The chemical structures were added, as requested.

  1. Ln 292, section 3.3 Please add molecular structure while discussing their biological properties

Answer: The chemical structure was added, as requested.

  1. References need to be adjusted according to the journal guideline.

Answer: The references were adjusted according to the journal guideline.

Round 2

Reviewer 1 Report

The authors included my comments sufficiently. I find this manuscript ready to be published.

Author Response

Thanks a lot